# Opinions towards Key Operational Aspects for the Implementation of HIV Self-Testing in Spain: A Comparison between Stakeholders and Potential Users

**DOI:** 10.3390/ijerph18041428

**Published:** 2021-02-03

**Authors:** Juan Hoyos, Juan-Miguel Guerras, Tomás Maté, Cristina Agustí, Laura Fernández-López, Luis de la Fuente, María-José Belza

**Affiliations:** 1Departamento de Salud Pública y Materno-Infantil, Universidad Complutense de Madrid, 28040 Madrid, Spain; hoyosmiller@hotmail.com; 2CIBER Epidemiología y Salud Pública (CIBERESP), 28029 Madrid, Spain; cagusti@iconcologia.net (C.A.); lflopez@iconcologia.net (L.F.-L.); lfuente@isciii.es (L.d.l.F.); mbelza@isciii.es (M.-J.B.); 3Centro Nacional de Epidemiología, Instituto de Salud Carlos III, Madrid 28029, Spain; 4Gerencia de Atención Primaria Valladolid Este, 47010 Valladolid, Spain; tomasmateenriquez@gmail.com; 5Centre Estudis Epidemiologics sobre les Infeccions de Transmissio Sexual i Sida de Catalunya (CEEISCAT), Departament de Salut, Generalitat de Catalunya, 08005 Badalona, Spain; 6Escuela Nacional de Sanidad, Instituto de Salud Carlos III, 28029 Madrid, Spain

**Keywords:** stakeholders, men who have sex with men, early diagnosis, self-testing

## Abstract

We assessed previous knowledge about the existence of HIV self-testing of stakeholders in Spain, as well as their personal position towards this methodology. We also assessed their views on potential users’ (PU) opinions towards several key operational aspects surrounding self-testing, and compared them to those expressed by a sample of PU comprised of men who have sex with men. In 2017, we recruited three types of stakeholders: public health professionals and policy makers (PHPPM) (*n* = 33), clinical providers (*n* = 290) and community-based/non-governmental organization (CBO/NGO) workers (*n* = 55). Data on PU (*n* = 3537) were collected in 2016. Previous knowledge about the existence of self-testing was higher in stakeholders than in PU, but being in favor was less frequent. PUs’ willingness to pay 25–30 euros for a self-test was higher than that which stakeholders considered. According to clinical providers and PHPPM, pharmacies would be PUs’ preferred place to obtain a self-test, which was in line with PUs’ actual choice. CBO/NGO workers on the other hand thought it would be CBO/NGOs. PHPPM and clinical providers considered primary care as PUs’ preferred setting to confirm a reactive self-test and CBO/NGO chose CBO/NGOs, but PUs preferred an HIV/STI testing service or clinic. Stakeholders’ opinions significantly differed from those of PUs. This divergence needs to be brought up to stakeholders as it could vary their position towards self-testing as well as the actions taken in the implementation of a testing option with the potential of increasing testing frequency.

## 1. Introduction

In Spain, estimations are that approximately 18% of people living with HIV remain undiagnosed [1]. Undiagnosed individuals could be facing barriers such as inadequate risk perception or fear of stigma and discrimination. They could also face barriers related to lack of anonymity or confidentiality, such as having to wait or having to return to receive the test result, and the need to discuss sexual behaviors [2].

Self-testing could contribute to the removal of some of the aforementioned barriers and facilitate access to testing. With HIV self-testing, individuals perform and interpret their own test. If reactive, they need to seek a confirmatory test. The potential advantages of self-testing are convenience, privacy, anonymity, and confidentiality. The concerns, on the other hand, have to do with the lack of face-to-face pre- and post-counselling, and with suboptimal linkage to the care of those obtaining a reactive self-test [3,4,5].

In 2016, the World Health Organization (WHO) issued recommendations on the incorporation of HIV self-testing as an additional testing method [6]. Since then, the strategy has spread across the world, and in Spain it was approved in January 2018 [7].

The knowledge, acceptability, and potential impact of self-testing has been evaluated mainly in men who have sex with men (MSM). MSM in Spain represented 56.4% of the new diagnoses of HIV reported in 2018 [8] and represent the group of potential users (PU) that could benefit the most from the introduction of self-testing. According to previous studies, this population has a relatively modest knowledge of the existence of self-testing [9] and are highly supportive of it [10,11]; a high percentage report that if testing had been available, they would have used it in the past. [10,11].

When introducing self-testing, the involvement of all relevant stakeholders is crucial for the success of the strategy at a national level. Public health professionals, policy makers (PHPPM), doctors and nurses working at healthcare settings (clinical providers), and community-based organization/non-governmental organization workers (CBO/NGO workers), all have a role to play in the design and implementation of this testing strategy, as well as in conducting advocacy activities to increase awareness and demand of HIV self-testing among PUs. However, their opinions towards self-testing have rarely been studied.

Some qualitative studies aimed at capturing key stakeholders’ opinions working in sub-Saharan countries [12,13] identified self-testing as an opportunity to increase testing. In Canada, a study that surveyed stakeholders engaged in HIV testing concluded that stakeholders were generally in favor of self-testing [14] but results were not provided by stakeholder type. Additionally, the difference between the opinions of stakeholders and PUs remain unknown.

In the era of expansion of HIV self-testing, our aim is to gain knowledge on the opinions regarding HIV self-testing of three different types of stakeholders involved in the design and implementation of HIV testing strategies in Spain. We do so by assessing their previous knowledge about the existence of self-testing, their personal opinion towards this testing option as well as their perception on what PUs opinions would be regarding several operational aspects surrounding the HIV-self testing process and compare them to those expressed by an actual sample of PUs comprising HIV-negative MSM.

## 2. Materials and Methods

Between 2016 and 2017, two online cross-sectional studies were conducted in Spain: one aimed at stakeholders, and the other aimed at MSM. Both were conducted in the context of the EUROHIVEDAT project (Operational knowledge to improve HIV early diagnosis and treatment among vulnerable groups in Europe) (Grant Agreement number 2013 11 01). The project counted with the approval of the ethical committee of investigation and animal welfare of the Instituto de Salud Carlos III (CEI PI 52_2015-v2) and the Hospital Germans Trias i Pujol (PI-14-106).

### 2.1. Study Participants

Three different types of stakeholders were recruited: PHPPM (*n* = 33), clinical providers (*n* = 290), and CBO/NGO professionals (*n* = 55). Participants were required to work in Spain.

Regarding the PUs, the inclusion criteria for recruitment were: ever having had sex with another man, currently living in Spain, meeting the minimum age required to legally have sex (16 years old), having been designated male at birth, and having self-reported being HIV negative. The sample of PUs was comprised of 3537 individuals.

### 2.2. Recruitment Procedures

The recruitment of stakeholders lasted from February to May 2017. Several professional associations, scientific societies, as well as advocacy and research groups were contacted, and all of them collaborated by sharing a link to the questionnaire with their members. Those who clicked on it were re-directed to an introductory screen where they were informed about the aim of the study, its anonymity, and funding sources. Before moving on to the first question, participants were required to give informed consent by checking a box with the following message: “I have read and understood the above information, and I want to participate”. Additionally, in person recruitment was carried out in the 2017 SEISIDA national conference. This conference is organized annually by the Spanish multidisciplinary AIDS society and is attended by stakeholders of the interest areas. Attendees were asked to participate by members of the research team. Those who agreed to participate were given a tablet to guarantee privacy. The initial screen showed the same introductory screen described above and the survey was exactly the same. No incentives were offered for participation.

The recruitment of PU was done mainly through gay geo-spatial “dating” applications and websites. Advertisement was done through banners, direct messages, and mailing lists. Those who clicked were directed to an introductory screen with information on the aims of the project, its anonymity, funding sources, and partners involved. Those who agreed to participate gave their informed consent by checking a box with the message “I have read and understood the above information, I am old enough to legally have sex, and I want to participate” before moving on to the first question. No incentives were offered for participation.

### 2.3. Data Collection Instrument

Data was collected through an online self-administered questionnaire. Instruments for stakeholders and PU were different, but had questions in common to enable comparison between groups.

For stakeholders, the questionnaire included questions to assess age and level of education and a section to assess their job profile. Before the section addressing self-testing, the following definition was included to ensure informed responses: “individuals take a sample, perform the test and obtain their result in under 20 min (just like pregnancy tests). There is no need of sending the sample to a laboratory as they are read and interpreted by the user”. Stakeholders were asked about their knowledge of the existence of self-testing (Yes/No) and about their personal position on this testing option. For the analysis, response options (in favor of, not sure, against of) were collapsed into a dichotomous variable (in favor of/not in favor of). Regardless of their personal opinion, they were asked to identify what they thought was the most important reason for the public to be against/in favor of self-testing.

Several key aspects for the implementation of HIV self-testing were also addressed. Firstly, stakeholders were asked if they thought that the public would be willing to pay the set price of 25–30 euros for a self-testing kit. The three original response options (“Yes”, “No, unless in great distress”, “Never”) were collapsed into a Yes/No variable for further analysis. Secondly, stakeholders’ opinions on what the public’s preferred setting would be to obtain a self-test and to confirm a reactive result were assessed. Preferred settings to obtain a self-test included the following: at supermarkets/drugstores; at community-based organizations or non-governmental organizations; through the internet; purchasing it by phone; at vending machines; only at pharmacies; other. Preferred settings to confirm a reactive self-test included the following: a hospital or clinic; HIV/STI testing service or clinic; the office of a medical specialist; general practitioner/family doctor; private laboratory; a pharmacy (where the test would be performed by a pharmacist); emergency room of a hospital; a mobile unit; community-based organization or non-governmental organizations; a bar/pub, club or sauna; elsewhere [Specify].

Finally, the potential use of HIV self-testing was also assessed. Potential use was assessed by asking stakeholders their opinion on whether the public would have used a self-test if already available. The initial five response options, “yes”, “probably yes”, “not sure”, “probably no”, and “no”, were transformed into a binary variable for analysis: “yes/probably yes” and “I am not sure/probably no/no”.

For PU, the questionnaire included a set of questions to assess sociodemography, sexual risk behaviors, and testing history. The section on self-testing was also introduced by a definition of the testing method and included the same questions as described in the stakeholders’ questionnaire, but in this case they all referred to their personal opinion.

### 2.4. Data Analysis

We first describe the general characteristics of stakeholders. The associations between variables were assessed using the Chi-square test or Fisher’s exact test when appropriate.

Prevalence ratios (PRs) and their corresponding 95% confidence intervals (CIs) were calculated by performing a Poisson regression analysis with robust variance to assess differences between the three different types of stakeholders and MSM on four different variables: personal opinion towards self-testing, knowledge about its existence, potential use if already available, and willingness to pay 25–30 euros for a self-test kit. Poisson regression was used because it is a better alternative than logistic regression in cross-sectional studies with frequent outcomes [15,16].

Stakeholders’ opinions on the reasons that the public could have to be in favor of and against HIV-self testing, and what the public’s preferred settings would be for purchasing and confirming a reactive result were explored and compared to opinions expressed by MSM.

## 3. Results

### 3.1. Stakeholders’ General Characteristics

Over 60% of the participants were PHPPM, 43.4% were clinical providers, and 20.4% of participants working in a CBO/NGO were ≥50 years of age (*p* = 0.006). Regarding educational level, virtually all PHPPM and all of the clinical providers had at least a university degree, whereas 27.8% of the participants working in a CBO/NGO had not finished a university degree at the moment of the survey (*p* < 0.001). Most of the PHPPM were HIV or public health officers/technicians (74.2%). Some 76.8% of the clinical providers worked in primary care and 80.3% were medical doctors. Regarding CBO/NGO workers, 32.7% reported working for an organization that targeted LGBT+ populations, 38.5% worked for CBO/NGOs that targeted other key populations, and 28.8% in one that did not target a specific population. Some 44.0% of the CBO/NGO participants reported that the activities of the organization they worked for were exclusively focused on HIV and other sexually transmitted infections (STI), 95.9% reported that it had an HIV testing and counselling service, and 73.6% that their job was directly involved with an HIV testing and counselling service (Table 1).

### 3.2. Knowledge and Opinions about Self-Testing

A total of 67.7% of the PHPPM knew about the existence of HIV self-testing, whereas this percentage was of 32.2% in clinical providers and 71.7% in CBO/NGO workers. Compared to PUs (14.5%), knowledge was higher among PHPPM (PR 4.7; CI95% 3.6–6.0), CBO/NGO workers (PR: 4.9; CI95%: 4.1–6.0), and clinical providers (PR: 2.2; CI95%:1.8–2.7) (Table 2).

The percentage that reported being in favor of self-testing was of 55.2% for PHPPM, 67.5% for clinical providers, and 35.3% for CBO/NGOs workers. Compared to PUs (86.9%), all three stakeholder groups were less favorable to HIV self-testing: PHPPM (PR 0.6; CI95% 0.5–0.9), CBO/NGO workers (PR: 0.4; CI95%: 0.3–0.6), and clinical providers (PR: 0.8; CI95%: 0.7–0.9) (Table 2).

### 3.3. Willingness to Pay and HIV Self-Testing Potential Use

The percentage that considered that the public would be willing to pay 25–30 euros was 37.5% in PHPPM, 29.4% in healthcare workers, and 17.6% in CBO/NGO. A total of 49.6% of the PUs reported that they would be willing to pay this price. Clinical providers (PR: 0.6; CI95%: 0.5–0.7) and CBO/NGO workers (PR: 0.4; CI95%:0.2–0.6) thought that the public’s willingness to pay the retail price for an HIV self-test was lower than what was expressed by PUs (Table 2).

Some 80.6% of the participant PHPPM reported that the public would have already used an HIV self-testing if already available. This percentage was of 73.3% in clinical providers and 60.8% for CBO/NGO workers. No statistically significant differences were found when compared to PUs (75.1%) (Table 2).

### 3.4. Reasons to Be Against and in Favor of Self-Testing

All three stakeholder groups thought that the presence of a trained person to counsel and inform about the result was the most important reason to be against self-testing for the public. This was especially true for PHPPM and for those working in CBO/NGOs (63.3% and 50.0%, respectively). The percentage for clinical providers was lower (38.1%) and similar to that of PU (36.4%). The second reason of importance for all stakeholders and PUs was that “obtaining the sample, performing the test and interpreting the results should be done by a trained professional” (PHPPM: 16.7%; healthcare workers: 22.6%; CBO/NGO workers: 23.9%; PU: 29.4%) (*p* = 0.066). (Table 3).

Privacy was considered to be, by all three stakeholder groups, the public’s most important reason to be in favor of self-testing (PHPPM: 41.4%; healthcare workers: 33.5%; CBO/NGO workers: 32.7%). This was also the case for PUs, but the percentage was lower (24.8%). Testing whenever they wanted was the second most frequently reported reason by PHPPM (17.2%) and PU (24.9%), but not for healthcare workers and CBO/NGOs who considered that avoiding judgmental attitudes was the second reason of importance for the public (26.4% and 22.4%, respectively) (*p* < 0.001) (Table 3).

### 3.5. Preferred Settings to Obtain a Self-Test and Confirm a Reactive Result

PHPPM (36.7%) and clinical providers (37.0%) considered that pharmacies were the preferred setting to obtain a self-test for the public. This appreciation was in agreement with the opinion expressed by PU (38.3%). Participants working in a CBO/NGO, on the other hand, considered that CBO/NGOs were the public’s preferred setting to obtain a self-test (33.3%). PHPPM (26.7%), workers of healthcare settings (22.2%), and PUs (23.9%) considered that the internet was the second most preferred setting. For CBO/NGO workers, pharmacies were considered as the second most preferred setting (26.7%). CBO/NGOs were considered as the third most relevant place to obtain a self-testing kit by PHPPM (23.3%) and by workers of healthcare settings (17.3%), but only by 4.3% of PUs (*p* < 0.001) (Table 4).

Preferred settings for confirmation purposes, varied by type of stakeholder. For PHPPM, general practitioners would be the public’s preferred choice (27.6%), followed by an HIV/STI testing service or clinics (24.1%), and hospitals and clinics (24.1%). This order was also followed by clinical providers, but the percentage that considered general practitioners as the preferred choice for the public was higher (55.3%). PUs also chose these three settings as their preferred choice for confirmation: an HIV/STI testing service or clinic was chosen by 27.7%, a general practitioner by 22.2%, and hospitals/clinics by 20.7%. The largest discrepancy was observed in the CBO/NGO workers who considered CBO/NGOs as the preferred option for confirmation for the public (39.1%), which was considered as a minority option by the other two stakeholder types and by PUs (*p* < 0.001) (Table 4).

## 4. Discussion

This is the first study conducted in high and middle-income countries that investigates stakeholders’ opinions about aspects related to HIV self-testing and presents results by type of stakeholder. We also compare their positions to those expressed by MSM who are one of the most relevant groups of PUs. HIV self-testing has proven to be a reliable testing option [17] and has shown its capacity to increase testing uptake under experimental conditions [18]. According to the European Centre for Disease Prevention and Control (ECDC), it needs to be made available to MSM in order to increase testing uptake and frequency of testing [19].

Our findings suggest that, in spite of having higher levels of prior knowledge, stakeholders were less supportive of self-testing and considered that PU willingness to pay the retail price for a self-test was lower than that reported by PUs. The largest differences were observed in CBO/NGO workers. Reasons to be in favor tended to converge across all groups, but reasons to be against self-testing and the preferred settings to obtain a kit and confirm a reactive result differed by type of stakeholder.

All participant stakeholders, but especially PHPPM and CBO/NGO workers, had a higher knowledge of the existence of HIV self-testing when compared to PUs, but the higher the knowledge the lower the probability of being in favor of this testing option. Participant stakeholders and PUs were given a standardized definition of self-testing at the beginning of the survey section that assessed it. This allowed everyone to give an informed response, but it is possible that having previous knowledge about the existence of this testing option could have led to deeper reflection on the limitations of self-testing and inclined stakeholders to less favorable opinions. However, this is a hypothesis that should be confirmed by further studies.

The opinion towards self-testing was especially unfavorable in CBO/NGO workers. As members of the civil society, CBO/NGOs are strong opinion leaders. From the beginning of the epidemic, they have being strong players in the HIV/AIDS advocacy field and have played a major role in supporting and pushing a number of important measures to fight against HIV [20], especially those working in the gay community who have a history of paving the road to other communities at heightened risk [21]. CBO/NGOs would be a strong ally in raising awareness and promoting HIV self-testing, especially if we take into account the low knowledge reported by PUs. Self-testing can be viewed as a competitive option to the rapid testing programs offered in nearly all of the CBO/NGOs that employed our participants, as they both could be targeting similar populations. The results of a study conducted in Canada pointed towards this direction, with stakeholders involved directly in testing activities viewing HIV self-testing as a challenge to current HIV testing delivery models [14]. However, self-testing is rapidly expanding across the world, and the WHO estimated that the global self-test volumes would grow from 1 million in 2017 to an estimated 16.4 million (12.9 million–19.3 million) by the end of 2020 [22]. In Spain, self-testing has just been incorporated into the national roster of testing options, offering an opportunity for CBO/NGOs to incorporate it into their services as a dispensation point, support line, or with programs aimed at linking to provide care for those who obtain a reactive result.

Clinical providers, on the other hand, reported lower prior knowledge about the existence of HIV self-testing, but they were quite supportive. This result is surprising if we consider that most of the participant clinical providers worked in primary care where professionals do not deal with HIV as frequently as in other settings such as sexual health clinics or HIV/STI units. Primary care professionals could consider HIV self-testing as an addition to a series of self-screening tests for health conditions that include home pregnancy tests and blood-glucose monitoring [23]. HIV self-testing could not only allow access to testing to individuals who perceive themselves at risk, but also alleviate the pressure on primary care.

Price is considered one of the main limitations when implementing self-testing as a nationwide strategy [24]. In Spain, the initial marketing price of a self-test was set to approximately 25–30 euros. Almost half of our PU reported they were willing to pay this amount, but this is not necessarily representative of all MSM, and other population groups could be less willing to accept this price. The rest of the stakeholders were less prone to considering that the public would be willing to pay this price. Again, this was especially true among CBO/NGO workers. This difference could be related to the characteristics of their target population. It is reasonable to think that higher willingness to pay 25–30 euros is associated to better economic status. However, prices could go down when kits from other manufacturers are incorporated into the market. Additionally, publicly funded programs could remove this barrier by partially or fully funding a nationwide program similar to what has been done in the UK with the HIV self-sampling service [25]. Regarding the potential use of self-testing, most of the participants in the three groups of stakeholders considered that if HIV self-testing had been available, it would have been used by PUs in the past. This opinion was shared by PUs and is very much in line with that expressed by MSM in other studies conducted in Spain [10,11].

All three stakeholder groups, as well as PUs, reported similar reasons to be against self-testing, but we identified differences between stakeholders’ and PUs’ reasons to be in favor. Thus, clinical providers and CBO/NGO workers thought that undergoing testing without having to face judgmental attitudes was an important reason to be in favor, but only a minority of PUs felt this way. The low importance given by PUs to this reason could mean that after 20 years of efforts towards normalizing HIV testing [26], we could have reached a point where MSM living in Spain might feel that judgmental attitudes towards their sexuality are no longer the barrier to access testing that it was before. This should be incorporated into the knowledge of frontline testers, such as the majority of clinical providers and CBO/NGO workers in our sample.

When implementing HIV self-testing, the dispensation points need to be well-thought-out. In this sense, stakeholders and PUs all agreed in considering pharmacies and the internet as relevant settings to obtain a self-test. On the other hand, stakeholders tended to overestimate the choice of CBO/NGOs when compared to PUs. Testing recommendations for PUs are that they should test at least yearly and up to every three months depending on ongoing risk, sexual behavior, or history of STI [19]. These are demanding recommendations, and MSM might value the convenience of obtaining a self-test in common and very frequented places such as the aforementioned over visiting the premises of a CBO/NGO.

Another key point to be considered is the preferred setting for confirming a reactive result. In this respect, all opinions were similar, except for CBO/NGO workers who considered their setting as the preferred option for PUs for confirmation purposes. However, confirmatory testing is almost never provided in CBO/NGOs as it is generally carried out by Western blot, a laboratory-based technology.

Sub-optimal linkage to care (understood as timely confirmation testing and ART initiation) is recognized as one of the main limitations of HIV self-testing, and WHO guidelines recommend the follow-up of those with a reactive test [6]. However, the strategies to support linkage to care have been insufficiently investigated [27].

The results are not without limitations. Data were collected before the approval of self-testing, and opinions towards it could have changed since then. Recently, the COVID-19 pandemic has implied severe lockdowns at a national level potentially shifting opinions towards more favorable positions towards distant testing methodologies such as HIV self-testing. In this sense, our data offers baseline information to compare against future studies. The generalization of our results is limited by the profile of PHPPM and clinical providers since they were mainly comprised of HIV/public health technicians and medical doctors working in primary care, respectively. The profile of CBO/NGO workers was more varied. To a certain extent, there could be a selection bias in our population since individuals with a strong opinion towards self-testing (in favor or against) were probably more likely to accept participation.

## 5. Conclusions

Evidence suggests that there have been great improvements in the reduction of the undiagnosed fraction of the epidemic [1]. For further reduction, new diagnostic options need to be incorporated. According to our data, stakeholders had a higher previous knowledge about the existence of HIV self-testing, but their views on the opinions and preferences of PUs, especially in CBO/NGO workers, differ from the ones expressed by the actual PUs. Stakeholders need to be informed about these divergences since they could translate into greater support for a new testing methodology that needs to be promoted through advocacy, health information activities, and new service delivery models in which they all play a relevant role.

## Figures and Tables

**Table 1 ijerph-18-01428-t001:** General characteristics of participant stakeholders.

	Public Health Professionals/Policy Makers (*n* = 33)		Clinical Providers (*n* = 290)		CBO/NGO Workers (*n* = 55)		*p*
	n		n	%	n	%	
**Age**							0.006
<30	1	3.0	35	12.1	8	14.8	
30–39	3	9.1	52	17.9	16	29.6	
40–49	9	27.3	77	26.6	19	35.2	
≥50	20	60.6	126	43.4	11	20.4	
**Education**							<0.001 *
3.0	1		0	0.0	15	27.8	
University degree	23	69.7	216	74.5	35	64.8	
Postgraduate	9	27.3	74	25.5	4	7.4	
**Job level**							
High senior official	2	6.5					
HIV and/or public health technician	23	74.2					
Other	6	19.4					
**Clinical setting**							
Primary care			215	76.8			
HIV/STI specific settings			37	13.2			
Secondary care setting (not HIV/STI specific)			16	5.7			
Other			12	4.3			
**Profession**							
Medical doctor			224	80.3			
Nurse			49	17.6			
Pharmacist			2	0.7			
Other			4	1.4			
**Target population**							
Mainly LGBT+ population					17	32.7	
Mainly other key populations					20	38.5	
Does not serve a specific group					15	28.8	
**Focus on HIV and/or other STIs**							
Exclusively focused on HIV/STI					22	44.0	
Although not exclusively it includes HIV/STIs							
**The CBO has an HIV testing counselling service**					47	95.9	
**Job directly involved with an HIV testing counselling service**					39	73.6	

CBO/NGO: Community-based organization/non-governmental organization; * *p* values corresponds to fishers exact test.

**Table 2 ijerph-18-01428-t002:** Stakeholders’ knowledge, opinion, intention to pay and potential use of HIV self-testing compared to that of potential users.

	Knows about the Existence of HIV Self-Testing			In Favor of HIV Self-Testing			Would Pay 25–30 Euros for an HIV Self-Test			Would Have Used an HIV Self-Test if Already Available		
Stakeholders (*n* = 378)	%	PR	95%CI	%	PR	95%CI	%	PR	95%CI	%	PR	95%CI
Public health professionals/policy makers (*n* = 33)	67.7	4.7	(3.6–6.0)	55.2	0.6	(0.5–0.9)	37.5	0.8	(0.5–1.2)	80.6	1.1	(0.9–1.3)
Clinical providers (*n* = 290)	32.2	2.2	(1.8–2.7)	67.5	0.8	(0.7–0.9)	29.4	0.6	(0.5–0.7)	73.3	1.0	(0.9–1.1)
CBO/NGO worker (*n* = 55)	71.7	4.9	(4.1–6.0)	35.3	0.4	(0.3–0.6)	17.6	0.4	(0.2–0.6)	60.8	0.8	(0.6–1.0)
**Potential users * (*n* = 3537)**	14.5	ref.		86.9	ref.		49.6	ref.		75.1	ref.	

PR: Prevalence ratio; 95%CI: 95% Confidence interval; CBO: Community-based organization; NGO: Non-governmental organization; * HIV-negative men who have sex with men.

**Table 3 ijerph-18-01428-t003:** Stakeholders’ opinion of reasons that the public has in favor/against self-testing vs. opinions of potential users.

	Stakeholders by Work Area (*n* = 378)						Potential Users * (*n* = 3537)		*p*
	Public Health Professionals/Policymakers (*n* = 33)		Clinical Providers (*n* = 290)		CBO/NGO Workers (*n* = 55)				
	*n*	%	*n*	%	*n*	%	*n*	%	
**Users’ reasons to be against self-testing ****									0.0066
The presence of an expert to counsel and inform about the result is essential	19	63.3	103	38.1	23	50.0	150	36.4	
Obtaining the sample, performing the test and interpreting the results should be done by a trained professional	5	16.7	61	22.6	11	23.9	121	29.4	
Concerns about the validity of the results	3	10.0	52	19.3	2	4.3	68	16.5	
Self-testing may help to maintain HIV as a matter of taboo/shame	2	6.7	25	9.3	4	8.7	33	8.0	
People could be forced to self-test in front of their partner	0	0.0	15	5.6	3	6.5	16	3.9	
Other	1	3.3	14	5.2	3	6.5	24	5.8	
**Users’ reasons to be in favor of self-testing ****									<0.0001
It helps to keep their privacy	12	41.4	90	33.5	16	32.7	782	24.8	
It helps to test whenever they can/want	5	17.2	54	20.1	7	14.3	785	24.9	
It saves time, paperwork, queues, waiting time	1	3.4	9	3.3	4	8.2	708	22.5	
It contributes to taking responsibility for their own health	3	10.3	23	8.6	3	6.1	499	15.8	
It helps to avoid intimate and personal questions	4	13.8	16	5.9	6	12.2	208	6.6	
It saves them judgmental attitudes (regarding sexual practices, sexual orientation…)	2	6.9	71	26.4	11	22.4	89	2.8	
It allows one to avoid counselling	0	0.0	2	0.7	1	2.0	10	0.3	
Other	2	6.9	4	1.5	1	2.0	70	2.2	

* HIV-negative men who have sex with men; CBO: Community-based organization; NGO: Non-governmental organization; ** For potential users this question as exclusive for those who reported being against/not sure of self-testing.

**Table 4 ijerph-18-01428-t004:** Stakeholders’ opinions on the public’s preferred setting to obtain a kit and to confirm a reactive result vs. opinions of potential users.

	Type of Stakeholders (*n* = 378)								*p*
	Public Health Professionals/Policymakers (*n* = 33)		Clinical Providers (*n* = 290)		CBO/NGO Workers (*n* = 55)		Potential Users * (*n* = 3537)		
	*n*	%	*n*	%	*n*	%	*n*	%	
**Preferred setting to obtain a self-testing kit**									<0.001
Pharmacies	11	36.7	94	37.0	12	26.7	701	38.3	
Through the internet	8	26.7	49	19.3	10	22.2	438	23.9	
Supermarkets/drugstores	2	6.7	35	13.8	5	11.1	378	20.7	
Vending machines	1	3.3	21	8.3	2	4.4	195	10.7	
Community-based/non-governmental organizations	7	23.3	44	17.3	15	33.3	78	4.3	
Purchasing it by phone	1	3.3	5	2.0	0	0.0	13	0.7	
Other	0	0.0	6	2.4	1	2.2	27	1.5	
**Preferred setting for confirmation**									<0.001
HIV/STI testing service or clinic	7	24.1	40	15.7	15	32.6	730	27.7	
General practitioner/family doctor	8	27.6	141	55.3	1	2.2	586	22.2	
Hospital or clinic (including the emergency room)	7	24.1	40	15.7	9	19.6	546	20.7	
Office of a medical specialist	4	13.8	23	9.0	3	6.5	510	19.3	
Community-based/non-governmental organizations	1	3.4	5	2.0	18	39.1	177	6.7	
Outreach activity (mobile unit, bar, club, or sauna)	1	3.4	0	0.0	0	0.0	14	0.5	
Others **	1	3.4	6	2.4	0	0.0	73	2.8	

* HIV-negative men who have sex with men; CBO: Community-based organization; NGO: Non-governmental organization; ** Includes pharmacies, private laboratories, and elsewhere.

## Data Availability

The datasets used and/or analyzed during the current study are available from the corresponding author on reasonable request.

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
