# Peer review of "Opinions towards Key Operational Aspects for the Implementation of HIV Self-Testing in Spain: A Comparison between Stakeholders and Potential Users"

_ijerph, 2021, doi:10.3390/ijerph18041428_

Round 1
Reviewer 1 Report
Overall, I think that this is an interesting paper and an important topic. There are issues, however, related to (a) the structure of the paper in terms of phrasing, (b) how information is presented, and (c) some of the conclusions that are made. I will not point out all of the issues, but provide some examples of each. Also, please have the manuscript reviewed by a native English speaker. If that was already done prior to submission, please have a second native English speaker review the resubmission.
Abstract:
Here and throughout the paper you use the terms "opinions" and "perspectives." These are pretty similar in meaning. Suggest you use only one.
Here and elsewhere your preposition use is awkward. For example, on Line 20 you have "comprised BY men who..." when it should be "comprised of men who...".
Make it clear that the PHPPMs, etc. listed in the 2nd sentence are examples of the stakeholders that you refer to in the 1st sentence.
Line 24: Add "of" after "favor" and make it clear what the 25-30 euros is for.
Line 27: Add "workers" after the 1st CBO/NGO.
Line 28: Confirmation setting of what?
Also, at the end of the Abstract a sentence or two about the implications of these findings would be ideal.
Introduction:
Line 35: Change "could be facing" to "could face barriers..."
Line 40: "...undiagnosed fraction of the epidemic" is awkwardly phrased.
Line 47: The specificity of "legally sold" suggests that previously there were illegal versions of HIV self-tests. Is that the case?
Line 53: Tense issues make this portion of the sentence a bit difficult to follow. It should be something like -- "...if testing had been available, they would have used it in the past."
Line 64: "are" should be changed to "were."
Line 64: Here and elsewhere you use the word "profile," which suggests a composition of traits that someone has. Suggest you change the word to "type" OR "group."
Line 69: What is meant by "operational aspects"?
Materials and Methods:
Line 81: Is this over their lifetime, in the past year in terms of engaging in same-sex activity?
Line 120: I believe this is the 1st mention of 25-30 euros, but it is not until the Conclusion that you note that this is the set price for such tests.
Line 123: What were the response options for the question about the public's preferred setting?
Line 135: I think the "of" between "test" and "Fisher" should be OR.
Results:
It should be Stakeholders' general characteristics
The general practice is to not start sentences with a number. Therefore, you could do something like:
Over 60% of the...
Also, "whereas this percentage was of..." is awkwardly phrased.
LGBT+, rather than LGTB+, is usually how this is phrased.
Lines 155-156: "...was devoted exclusively to HIV" is a bit awkwardly phrased. Are they devoted to treating HIV, preventing HIV...
Line 161: Change "positions towards" to "opinions about."
Line 186: "All three stakeholders" suggests that there were a total of 3 people who were surveyed. Change to "All three stakeholder GROUPS..."
Line 203 and elsewhere -- don't put a period after the p-value AND after the table information. A period after the table (Table 3) is sufficient.
Discussion:
Make the current 2nd paragraph the 1st paragraph of the Discussion.
Line 228: Knowledge and opinions about what? The opening sentence is too general.
Line 243-245: This sentence is not supported by the data. You don't know that previous knowledge leads to a greater appreciation for limitations.
Line 257: "...apparently here to stay" is too informal for a research paper.
Line 282-284: First, paragraphs are usually more than one sentence. Second, I am not sure how useful it is to know stakeholders' (or anyone's) opinion about what a group may or may not have done had self-testing been previously available. Surely, it is more important to know what people will do now that such testing exists.
Lines 293-295: The first full sentence has a number of problems. As phrased, it suggests that it is the MSM who have worked for 20 years. Even if you are referring to the CBO/NGO workers, not all of them would have worked in the field for 20 years. The full sentence on Line 295 is unclear and awkwardly phrased.
Line 298: Based on the data, you can't conclude that stakeholders overestimated the public's preferences. You only asked stakeholders and a group of potential users. The public would be a different sample, which could include, but would not be limited to, potential users.
The bigger issue is that in the end there are at least four populations (three groups of stakeholders and potential users) who have different views about self-testing. And it's unclear what the main takeaway message is. For example, if the three stakeholder groups have slightly different levels of support for self-testing, does that mean that outreach to these groups is warranted? Basically, what do we do with this information? How is it useful to those who want to increase testing and HIV status knowledge?
Reviewer 2 Report
Great article. I have a few grammar items and potential rewordings, but otherwise it's really well done, no major data issues from me.
228: differed
228: lose comma after stakeholders
228: I would actually put the first paragraph (Knowledge...) after the third paragraph (All participant...), and start the discussion with the 2nd paragraph (This is...)
241: drop "was" after lower
239: This third paragraph probably needs a polishing for grammar and clarity
256: the sentence "But self testing..." is too casual. I get the point but try moving that idea to the start of the next sentence. Maybe just drop the sentence and add "however" to the start of the next sentence
259: "In Spain, self-testing has..."
260: I would replace "a good" with "an"
266: frequent'ly', and no comma after professionals
274: comma after MSM
282: feels out of place, I would move this into one of the above paragraphs
285: comma after stakeholders and PU
285: remove "of"
286: replace "a gap" with "a difference" or something similar, it was unclear
295: that "it" was before
295: the last sentence can be dropped or at least reworded and added to the previous sentence
296: this paragraph is fine but doesn't seem very necessary
304: replace "To" with "In"
307: "by" Western Blot
310: Make "but the strategies" its own sentence
312: these last three lines are unnecessarily wordy
318: quickly define those more favorable positions
329: no comma after stakeholders
331: comma after cascade
Reviewer 3 Report
This is a sound original study contributing to our knowledge about self-testing in two different categories of population, that of service providers and potential users.
The only suggestion this reviewer has is to elaborate discussion on the reliability of self-testing and evidence about its use and role in prevention of HIV.
This is a well-written manuscript and reader friendly.
Thank you for the opportunity to review.
